# FLOW-OF-ACTION: SOP ENHANCED LLM-BASED MULTI-AGENT SYSTEM FOR ROOT CAUSE ANALYSIS

## ABSTRACT

In the realm of microservices architecture, the occurrence of frequent incidents necessitates the employment of Root Cause Analysis (RCA) for swift issue resolution. It is common that a serious incident can take several domain experts hours to identify the root cause. Consequently, a contemporary trend involves harnessing Large Language Models (LLMs) as automated agents for RCA. Though the recent ReAct framework aligns well with the Site Reliability Engineers (SREs) for its thought-action-observation paradigm, its hallucinations often lead to irrelevant actions and directly affect subsequent results. Additionally, the complex and variable clues of the incident can overwhelm the model one step further. To confront these challenges, we propose **Flow-of-Action**, a pioneering Standard Operation Procedure (SOP) enhanced LLM-based multi-agent system. By explicitly summarizing the diagnosis steps of SREs, SOP imposes constraints on LLMs at crucial junctures, guiding the RCA process towards the correct trajectory. To facilitate the rational and effective utilization of SOPs, we design an SOP-centric framework called **SOP flow**. SOP flow contains a series of tools, including one for finding relevant SOPs for incidents, another for automatically generating SOPs for incidents without relevant ones, and a tool for converting SOPs into code. This significantly alleviates the hallucination issues of ReAct in RCA tasks. We also design multiple auxiliary agents to assist the main agent by removing useless noise, narrowing the search space, and informing the main agent whether the RCA procedure can stop. Compared to the ReAct method's 35.50% accuracy, our Flow-of-Action method achieves 64.01%, meeting the accuracy requirements for RCA in real-world systems.

## 1 INTRODUCTION

Traditional monolithic applications encounter notable challenges including intricate deployment processes and limited scalability, attributed to the proliferation of services and frequent service iterations. In response to this context, Microservices Architecture (MSA) has surfaced and continually evolved (Chen et al., 2024a). By disassembling monolithic applications into small, self-sufficient service units, each dedicated to specific business functionalities, MSA presents benefits such as loose coupling, independent deployment, and effortless scalability. Nevertheless, with the escalation of user numbers and their corresponding demands, the diversity and quantity of MSA instances also increase. Despite the implementation of numerous monitor tools, recurrent incidents arise from hardware malfunctions or misconfigurations, posing challenges to reliability assurance. These incidents lead to substantial financial losses. For instance, on November 12, 2023, Alibaba experienced a large-scale outage, resulting in the interruption of multiple services for nearly three hours[1].

To promptly tackle these incidents, Root Cause Analysis (RCA) has emerged as a prominent research area within Artificial Intelligence for IT Operations (AIOps) in recent years. Traditional RCA techniques, in order to address the difficulties of manual fault diagnosis, have employed deep learning methods to learn from historical faults (Li et al., 2022b). However, these methods have two main drawbacks. First, they have poor adaptability to new scenarios, requiring model retraining when faced with a new situation. Second, they only output the root cause of the fault without providing the entire diagnostic process, resulting in poor explainability. This situation often results

---

[1]https://www.datacenterdynamics.com/en/news/alibaba-cloud-hit-by-outage-second-in-a-month/

in Site Reliability Engineers (SREs) harboring a sense of distrust towards the results, as they fear that misidentifying the root cause could potentially result in further wasted repair time or exacerbate faults by addressing the wrong issue. Over the recent years, Large Language Model (LLM) agents like ReAct (Yao et al., 2022) and ToolFormer (Schick et al., 2024) have been deployed across diverse domains. LLM agents harness their robust natural language understanding capabilities to adeptly coordinate various tools, allowing SREs to see the entire troubleshooting process and providing rich explanations for the root causes. Nonetheless, despite the considerable prowess of LLM agents, the efficient and accurate utilization of LLM agents in RCA encounters ongoing challenges.

**Challenge 1: Randomness and hallucinations leading to irrational action selection**

Current LLMs primarily function as probabilistic models (Radford, 2018; Radford et al., 2019), thereby exhibiting pronounced randomness and tendencies towards generating hallucinations. Employing an LLM agent for RCA activities necessitates the retrieval and comprehension of diverse data modalities (metric (Misiakos et al., 2024), log (Rosenberg & Moonen, 2020), trace (Yao et al., 2024b)) and the extensive utilization of API tools. As the scope of the context expands, issues often emerge such as inaccurate parameter extraction leading to failures in tool invocation and discrepancies between tool invocations and the context at hand. Instances of randomness or hallucinations at any stage can significantly impact the subsequent trajectory of the RCA procedure, hindering the accurate identification of the true root cause.

**Challenge 2: Complex and variable observations leading to multiple reasonable actions**

Existing LLM agents are typically bundled with a diverse array of tools (Qin et al., 2023), especially within complex domains like RCA, where the number of APIs can escalate to hundreds. Each API invocation results in varied observations, thereby

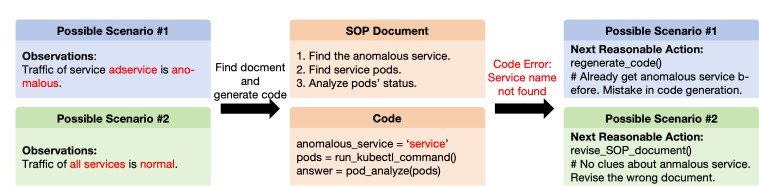

Figure 1: Illustration example of challenge 2.

introducing intricacies in action selection. Furthermore, even when confronted with identical observations, multiple plausible actions may be viable. For example, as shown in Figure 1, within the context of a code error "Service name not found", the root cause could originate from errors in the code generation phase or inaccuracies in associated SOP document, prompting multiple feasible actions like code regeneration or document revision.

To confront the challenges outlined above, we propose **Flow-of-Action**, a Standard Operating Procedure (SOP) enhanced Multi-Agent System (MAS). Initially, to mitigate the impact of randomness and hallucinations in the orchestration process, we integrate SOPs into the knowledge base and propose the **SOP flow**. Specifically, SOPs outline a standardized set of steps for RCA, while SOP flow represents an efficient and accurate process built upon SOPs for their effective utilization. Through prompt engineering, we ensure that the orchestration of the main agent loosely follows the SOP flow in the absence of unexpected circumstances. Subsequently, to tackle the second challenge, compared with the thought-action-observation paradigm, we propose the thought-**actionset**-action-observation paradigm. Flow-of-Action avoids immediate action selection and instead generates a reasoned action set before making the final decision on the course of action. Besides, we devise a novel MAS. Specifically, we introduce multiple agents such as MainAgent, CodeAgent, JudgeAgent, ObAgent, and ActionAgent, each entrusted with distinct responsibilities, collaborating harmoniously to enhance root cause identification.

Our key contributions are summarized as follows:

- We propose the Flow-of-Action framework, the first agent-based fault localization process centered around SOPs. With this framework, we significantly reduce the inefficiency in action selection of the native ReAct framework and reducing the cost of trial and error.

- We introduce the concept of SOPs to integrate the expert experience into the LLM to greatly reduce hallucinations during RCA. For any given fault, we can automatically match the

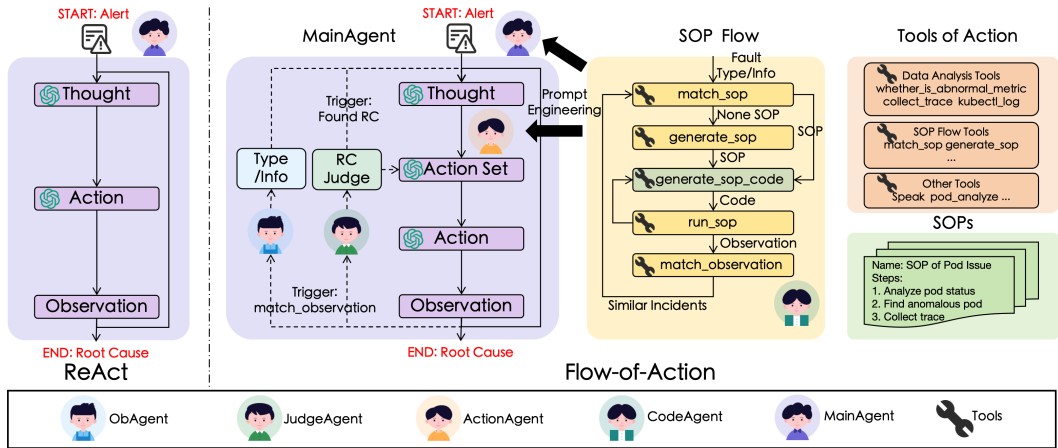

Figure 2: Comparison of ReAct and Flow-of-Action. RC means root cause. Dashed lines represent paths triggered under specific conditions. When the previous action is $match\_observation$, JudgeAgent and ObAgent are triggered. When JudgeAgent finds the root cause, it triggers the input of the analysis result to thought and adds $Speak$ to action set.

> most relevant set of SOPs and can also generate new SOPs automatically, extending the limited set of human-generated SOPs.

- We innovatively propose a multi-agent collaborative system, including JudgeAgent and ObAgent. JudgeAgent assists the MainAgent in determining whether the root cause of the fault has been identified in the current iteration, while ObAgent helps MainAgent extract fault types and key information from massive amounts of data, addressing the information overload issue in the RCA process.

- Through a fault-injection simulation platform of a real-world e-commerce system, Flow-of-Action has increased the localization accuracy from 35% to 64% compared to ReAct, proving the effectiveness of the Flow-of-Action framework.

## 2 FLOW-OF-ACTION

In this section, we will present the design of Flow-of-Action. As illustrated in Figure 2, the Flow-of-Action is a MAS built upon the ReAct. It encompasses three key design components: the SOP flow, the action set, and the MAS. We will delve into each of these components in the subsequent sections. Prior to their detailed exploration, we will introduce the foundational knowledge required, including the knowledge base and tools utilized by the Flow-of-Action.

### 2.1 KNOWLEDGE BASE OF AGENTS

Given the restricted context length of LLMs, Retrieval-Augmented Generation (RAG) has experienced notable progress (Jeong et al., 2024). However, the quality of text retrieved by RAG significantly influences the ultimate outcomes. Many existing RAG methodologies segment documents within the knowledge base and employ semantic block embeddings to calculate similarity for retrieval. This approach, however, does not consistently yield optimal results in RCA. Therefore, we have devised an innovative knowledge base model integrating SOP knowledge and historical incident knowledge.

#### 2.1.1 SOP KNOWLEDGE

With the successful integration of SOPs in the realm of code generation (Hong et al., 2023), there is a growing recognition that relying solely on LLMs to execute intricate tasks like RCA is impractical. SOPs, to a certain extent, impose constraints on LLMs at crucial junctures, guiding the entire process towards the correct trajectory. Consequently, we have embedded SOPs into the knowledge base,

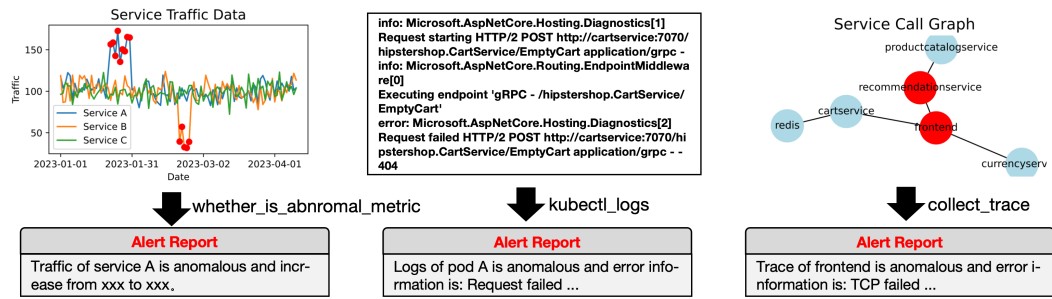

Figure 3: Multimodal data collection and analysis.

which are either authored by engineers based on domain expertise or extracted through automation tools. As shown in Figure 2, each SOP constitutes a self-contained unit comprising two attributes: name and steps. The name encapsulates essential information about the SOP, which is translated into a vector for subsequent retrieval purposes.

### 2.1.2 HISTORICAL INCIDENTS

As highlighted by Chen et al. (2024b), in systems where similar incidents occur frequently, historical incident data proves invaluable in identifying the root cause of ongoing incidents. Consequently, we incorporate the performance details of historical incidents into the knowledge base. Each historical incident is characterized by two key attributes: manifestation and type. When retrieving similar incidents, we evaluate similarity by comparing the embedding of the current observation with the embedding of the manifestation of historical incidents. However, relying solely on embeddings for assessment can introduce significant errors. To tackle this issue, we have intentionally devised the ObAgent (elaborated upon subsequently) to address this challenge.

### 2.2 TOOLS OF AGENTS

Within LLM agents, tools typically refer to pre-defined functions. During the action phase, LLM invokes relevant tools to obtain the necessary information. In Flow-of-Action, the tools utilized primarily fall into three categories: tools for multimodal data collection and analysis, tools related to SOP flow, and other tools. Each category will be discussed in detail below.

### 2.2.1 MULTIMODAL DATA COLLECTION AND ANALYSIS

Within the realm of MSA, which encompasses diverse modalities of data such as metrics, traces, and logs, the importance of multimodal data for RCA has been underscored by existing methodologies (Yao et al., 2024a; Yu et al., 2023). Consequently, we have implemented a comprehensive monitoring system to aggregate multimodal data. While LLMs excel in processing textual data, their effectiveness in interpreting structured data types like metrics is constrained, especially in the presence of data noise. Therefore, it is imperative to preprocess the data by denoising and transforming it into textual format for enhanced comprehension by LLMs. As depicted in Figure 3, we have devised the following components: $whether\_is\_abnormal\_metric$ to leverage time series anomaly detection algorithms (Wang et al., 2024) for identifying metric anomalies and converting them into fault-related text; $collect\_trace$ for capturing abnormal span details across the entire call chain and converting them into text format; and $kubectl\_logs$ for extracting abnormal log information from each pod within the Kubernetes system.

### 2.2.2 SOP FLOW TOOLS

As previously mentioned, we have introduced a flow centered around SOPs. This comprehensive flow is meticulously crafted based on common workflows employed by SREs in practical settings, integrating innovative concepts such as code. Details regarding the tools utilized within the flow are delineated in Table 1. Moreover, to preempt unexpected incidents during the flow's operation, we have developed a variety of targeted auxiliary tools. For example, within the context of

Table 1: Description of SOP flow tools.

|  | Input | Output | LLM Usage |
|---|---|---|---|
| match_sop | Fault Type/Information | SOP | No |
| generate_sop | Fault Type/Information | SOP | Yes |
| generate_sop_code | SOP | SOP Code | Yes |
| run_sop | SOP Code | Result after running the code | No |
| match_observation | Observation | Similar incidents | No |

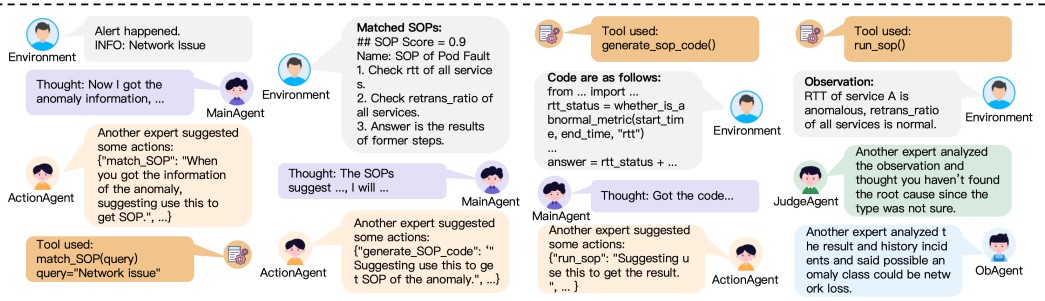

Figure 4: Example of Flow-of-Action.

$generate\_sop$, we have introduced $get\_relevant\_metric$ to streamline the retrieval of pertinent metric names.

### 2.2.3 OTHER TOOLS

The flow aims to establish a standardized and generalized process for intricate RCA tasks, devoid of service- or business-specific components within the tools themselves. However, a broader array of tools is necessitated when generating SOPs or SOP code, or when executing operations beyond the flow, to query the authentic operational state of the system. In addition to the previously mentioned tools for querying and analyzing multimodal data, a suite of tailored analysis tools has been devised for MSA, including $pod\_analyze$ and $service\_analyze$. These tools employ queries on specific attribute data within the Kubernetes system to ascertain the system's status. Upon identification, $Speak$ is employed to communicate the discovered root cause to all pertinent stakeholders. For a comprehensive elucidation of these tools, kindly consult the appendix.

### 2.3 SOP FLOW

The SOP flow represents a comprehensive logic chain of actions tailored to the SOP mentioned earlier. It serves to instruct LLMs on how to effectively utilize SOP knowledge. For instance, in the initial stages of RCA, it is essential to identify which SOPs are most relevant to the incident (corresponding to $match\_sop$). Additionally, if a particular incident does not align with any existing SOP, the automation of SOP generation should be considered (corresponding to $generate\_sop$). While the comprehensive SOP flow can be visually represented, as illustrated in Figure 2, in practical application, the full SOP flow is presented in the form of prompts to the MainAgent to aid in thought processes and to the ActionAgent to generate a more rational action set. By implementing such soft constraints, we aim to tackle the issue of chaotic tool orchestration while still maintaining the flexibility of LLMs. Unlike methods like FastGPT (Labring, 2023), we do not enforce strict workflow constraints on LLM orchestration. Figure 9 provides an example of the Flow-of-Action. Subsequently, we will systematically elucidate critical transitional subflows within the SOP flow.

#### 2.3.1 FAULT TYPE/INFORMATION→SOP

In our flow, we initially utilize $match\_sop$ to associate the fault information with the relevant SOP. This matching process involves computing the similarity between the current query and all SOP name embeddings, ranking them, and selecting the top $k$ matches. To avoid matching with highly

irrelevant SOPs, a filtering threshold is established. Nevertheless, in real-world contexts where new fault types frequently emerge, instances may arise where pertinent SOPs cannot be matched. To tackle this challenge, we introduce $generate\_sop$ to devise new SOPs for queries that do not align with existing SOPs. Specifically, we utilize LLMs to generate new SOPs and leverage existing SOPs as few-shot prompts to guide the development of more standardized and coherent SOPs.

Within the entirety of the flow, the generation of SOPs stands as a pivotal phase as it directly influences the subsequent RCA process. To enhance the precision of RCA, we have devised hierarchical SOPs. Our objective is for the RCA process to progress from a macro to micro level, from a general to specific perspective, mirroring real-world scenarios more closely. For instance, we first address network issues before delving into network partition problems.

### 2.3.2 SOP→SOP CODE

Once a suitable SOP is obtained, due to the interdependence of steps within the SOP, it is generally necessary to execute the SOP step by step to achieve the desired outcome. However, in real-world scenarios, SOPs are typically concise texts, making it relatively difficult for engineers lacking domain knowledge to execute the entire SOP. Utilizing an agent based on LLM to execute the SOP is a more rational and efficient approach. However, directly instructing the agent to execute all steps of the SOP one by one often leads to errors. This is because LLM tends to focus more on proximal text, and the outcome of a particular step can significantly influence the selection of subsequent actions.

Therefore, we have designed $generate\_sop\_code$ to convert the entire SOP into code for simultaneous execution. This approach offers three main advantages. Firstly, numerous works, including Chain-of-Code (Li et al., 2023), have demonstrated that executing code in LLM environments is far more accurate than executing text (Pan et al., 2023), aligning well with the precise requirements of RCA. Secondly, in many scenarios, including RCA, there exist numerous atomic operations where we wish for several actions to be executed together or none at all, as executing a single action in isolation may not yield useful results. SOPs exemplify this situation, where executing only a portion may not yield the desired fault information. Converting SOPs to code effectively addresses this issue, as once the code is executed, it must run from start to finish. Lastly, SOP code represents a collection of multiple actions, enabling the execution of multiple actions with a single tool invocation, thereby significantly reducing LLM token and resource consumption.

### 2.3.3 SOP CODE→OBSERVATION

After obtaining the SOP code, the flow invokes $run\_sop$ to execute the entire SOP code. However, the generation of code is not always accurate and may lead to various issues, such as syntax errors or incorrect variables within the code. In such instances, our flow expects to re-match suitable parameters and use $generate\_sop\_code$ to generate new, correct code. Once the code is error-free, we can smoothly execute it to obtain the desired results.

### 2.3.4 SOP CODE→FAULT TYPE/INFORMATION

As mentioned earlier, the definition of SOP is hierarchical, and our RCA process follows a layered and progressive approach. Upon executing $run\_sop$ and obtaining a new observation, we seek guidance to determine the next steps in the localization process. The ideal approach is to identify potential fault types based on the observation. Relying solely on the domain knowledge of the LLM agent is evidently insufficient for accurate judgment in a specific domain, necessitating fine-tuning of the LLM model or the introduction of more domain-specific knowledge. Inspiration from various methods (Chen et al., 2024b) suggests that most fault types have occurred historically. Therefore, we use $match\_observation$ to recall similar historical incidents based on observation. The ObAgent is then utilized to determine potential fault types or provide descriptions of faults for subsequent RCA processes.

### 2.4 ACTION SET

In section 1, we mentioned that in RCA, it is relatively challenging for the LLM agent to perform reasonable planning. This difficulty primarily arises from two reasons: the variability of observations and the existence of multiple possible actions for a given observation. Instantaneously iden-

tifying and executing the most reasonable action from numerous viable choices is an exceedingly challenging task for the LLM.

To address this challenge, we have devised a mechanism known as the action set. Specifically, drawing inspiration from the CoT (Wei et al., 2022), we first generate a series of reasonable actions comprising a set, with each action accompanied by a textual explanation of the rationale behind its selection. This set primarily consists of two components: actions generated by the ActionAgent and actions identified by the JudgeAgent. The ActionAgent incorporates flow information and numerous examples in the prompt to enhance the rationality of the generated actions. However, this may still overlook reasonable flow actions. Therefore, we have established a rule based on the flow to ensure that the action set is comprehensive and logical. For instance, if the preceding action was $generate\_sop$, the subsequent action of $generate\_sop\_code$ is added to the set. Secondly, the JudgeAgent evaluates whether the root cause has been identified during the current RCA process. If the root cause is pinpointed, the action $Speak$ is included in the action set.

Through action set, we have effectively mitigated the challenges posed by diverse observations and a plethora of feasible actions that could potentially hinder agent planning. Furthermore, the strategic design of the action set has enabled the LLM Agent to attain a nuanced equilibrium between stochasticity and determinism. Within RCA, excessive randomness may induce divergence in the localization process, impeding the formation of effective diagnostics. Conversely, an overly deterministic approach may incline the model towards scripted operations, limiting its capacity to handle unforeseeable and rapidly changing circumstances.

### 2.5 MULTI-AGENT SYSTEM

We have designed a MAS consisting of a single main agent along with multiple auxiliary agents. The MainAgent serves as the principal entity with authority, while the other agents are responsible for providing suggestions to it. The MainAgent orchestrates the entire localization process. The ActionAgent provides a feasible set of actions for the MainAgent to choose from. The ObAgent offers potential anomaly types or information after the MainAgent completes $match\_observation$. The JudgeAgent determines whether the root cause has been identified. However, even if the JudgeAgent believes the root cause has been found, the MainAgent may not necessarily use $Speak$ to conclude the entire localization process. Taking additional steps and gathering more information may lead to a more accurate root cause determination. The CodeAgent plays a crucial role in the SOP flow, possessing information on all tools and generating appropriate code for subsequent use. Through the MAS, the burden on the MainAgent is significantly reduced. It only needs to consider the opinions of other agents and make relatively accurate judgments based on the entire localization process. Such division of labor also aligns more closely with real-world operational scenarios.

## 3 EVALUATION

### 3.1 EXPERIMENT SETUP

#### 3.1.1 DATASET

We have deployed the widely used microservices system GoogleOnlineBoutique[2], an e-commerce system consisting of over 10 services, on the Kubernetes platform. Building upon this, we have implemented Prometheus, Elastic, DeepFlow, and Jaeger to collect metric, log, and trace data (Detailed in Appendix B.2). Anomalies are injected into microservices' pods using ChaosMesh[3]. There are a total of 9 types of anomalies injected, including CPU stress and memory stress (detailed in Table 5). Leveraging this setup, we have generated a dataset comprising 90 incidents. Further elaboration on these details can be found in the appendix.

#### 3.1.2 EVALUATION METRIC AND BASELINE METHODS

In the field of RCA, the specific location of the root cause is a critical focus for SREs. Additionally, categorizing the type of root cause is equally important, as SREs often specialize in different de-

---

[2]https://github.com/GoogleCloudPlatform/microservices-demo
[3]https://github.com/chaos-mesh/chaos-mesh

Table 2: Performance of different models. The best scores for each evaluation metric are bolded, and the second-best scores are underlined. Exclusive utilization of the APL metric is restricted to methodologies leveraging LLM agents. The fixed and specific accuracy of K8SGPT and Holmes-GPT, i.e. 11.11, is due to their ability to handle only one type of fault.

| Model | Base | LA | TA | Average | APL |
|---|---|---|---|---|---|
| K8SGPT | GPT-3.5-Turbo | 11.11 | 11.11 | 11.11 | - |
| HolmesGPT | GPT-3.5-Turbo | 11.11 | 11.11 | 11.11 | - |
| CoT | GPT-3.5-Turbo | 20.89 | 15.56 | 18.26 | - |
| CoT | GPT-4-Turbo | 36.00 | 29.22 | 32.61 | - |
| ReAct | GPT-3.5-Turbo | 13.11 | 25.22 | 19.17 | **9.41** |
| ReAct | GPT-4-Turbo | 47.67 | 23.33 | 35.50 | 10.76 |
| Reflexion | GPT-3.5-Turbo | 21.56 | 22.22 | 21.89 | 22.38 |
| Reflexion | GPT-4-Turbo | 33.67 | 24.44 | 29.06 | 28.09 |
| Flow-of-Action | GPT-3.5-Turbo | 54.22 | 53.89 | 54.06 | 18.83 |
| Flow-of-Action | GPT-4-Turbo | **70.89** | **57.12** | **64.01** | 15.10 |

partment like networking group or hardware group. Therefore, we have designed evaluation metrics focusing on both root cause location and fault type. Following the principle from mABC (Zhang et al., 2024), we consider redundant causes to be less detrimental than missing causes. Hence, we utilize two metrics: Root Cause Location Accuracy (LA) and Root Cause Type Accuracy (TA).

$$LA = \frac{L_c - \sigma \times L_i}{L_t}, TA = \frac{T_c - \sigma * T_i}{T_t} \qquad (1)$$

$L_c$ and $T_c$ represent all correctly identified root cause locations and types, while $L_i$ and $T_i$ denote the incorrectly identified locations and types. $L_t$ and $T_t$ represent total number of locations and types. $\sigma$ serves as a hyperparameter with a default value of 0.1. To prevent an excessive number of root causes, we limit the maximum number of root causes to three in LLM-based methods. In addition, we employed the Average Path Length (APL) to evaluate the efficiency of the LLM Agents. APL is defined as $\frac{\sum_{k=1}^{N} L_k}{N}$, where $L_k$ represents the diagnosis path length of the k-th sample, and $N$ denotes the number of samples for which diagnosis was completed within the specified maximum path length.

Regarding baseline methods, we have chosen several open-source Kubernetes RCA tools, such as K8SGPT (k8sgpt ai, 2023) and HolmesGPT (robusta dev, 2024). Since the implementation of RCA agents is highly specific to the scenarios, they are not open-source and are challenging to migrate. Therefore, we have developed some general-purpose open-source frameworks, such as CoT (Wei et al., 2022), ReAct (Yao et al., 2022), and Reflexion (Shinn et al., 2024), to serve as our baselines.

## 3.2 RQ1: OVERALL PERFORMANCE

Based on Table 2, our Flow-of-Action surpasses the SOTA by 23% in the LA metric and 28% in the TA metric. Despite the support of LLMs, K8SGPT and HolmesGPT continue to exhibit poor performance. This can be attributed to the significant limitations in the information they access. For instance, K8SGPT primarily queries Kubernetes metadata for attribute information, which is often insufficient for RCA, as faults may not necessarily manifest in metadata. CoT performs reasonably well in some common simple tasks due to the robust reasoning capabilities of LLMs. However, in RCA, where tasks are complex and diverse scenarios arise, even seasoned SREs struggle to promptly determine a series of pinpointing steps. Consequently, CoT fares poorly in the RCA domain. While ReAct integrates reasoning for each observation, the array of tools and diverse observations present challenges in rational orchestration. This is why we introduce the action set and SOP flow. Reflexion builds upon ReAct by introducing a path reflection mechanism. However, given that previous paths are predominantly incorrect, reflecting on a wealth of erroneous knowledge makes it arduous to arrive at accurate insights.

In terms of the APL metric, ReAct often erroneously identifies root causes due to a lack of proper judgment criteria, resulting in a relatively low APL. In contrast, Reflexion necessitates continuous

Table 3: Ablation study. The LLM backbone we use is GPT-3.5-Turbo.

| Method | LA | TA | Average | APL |
|---|---|---|---|---|
| **Flow-of-Action** | **54.22** | **53.89** | **54.06** | 18.83 |
| w/o SOP Knowledge | 8.56 | 22.11 | 15.39 | 20.00 |
| w/o SOP Flow | 15.11 | 39.89 | 27.50 | 19.78 |
| w/o Action Set | 44.67 | 40.00 | 42.34 | **11.48** |
| w/o ActionAgent | 32.78 | 34.56 | 33.67 | 18.42 |
| w/o ObAgent | 40.11 | 28.67 | 34.39 | 19.31 |
| w/o JudgeAgent | 36.11 | 33.89 | 35.00 | 20.00 |

path reflection, leading to numerous iterations and a higher APL. Flow-of-Action maintains an APL within an acceptable range, crucial for optimal performance in RCA tasks. In RCA tasks, the APL's magnitude is not fixed. Excessive values can escalate resource consumption and induce knowledge clutter, while inadequate values may lead to incomplete knowledge.

## 3.3 RQ2: IMPACT OF ACTION SET SIZE

As shown in Figure 2, we have introduced the action set mechanism, where the size of the action set impacts the subsequent selection of actions. We conducted validation on a subset of the dataset and the results are shown in Figure 5. We observed that the LA and TA remain relatively stable with changes in the action set size. This stability is attributed to the fact that, despite variations in the action set size, relevant flow tools are encompassed within the action set due to the constraints of the rules in SOP flow. Furthermore, the entire RCA process typically follows the flow, thereby minimizing significant fluctuations in accuracy. However, as the size increases, accuracy initially rises and then declines. This phenomenon occurs because smaller action sets restrict randomness, rendering the

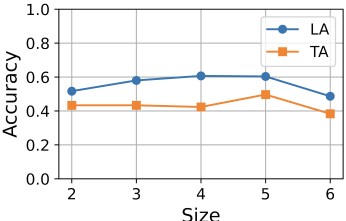

Figure 5: Accuracy of different action set sizes.

model incapable of handling complex scenarios. Conversely, larger sizes introduce more randomness, leading to a loss of control by the model. Hence, we opt for a moderately sized default value of 5 as it strikes a balance between these extremes.

## 3.4 RQ3: ABLATION STUDY

We conducted a detailed ablation study by removing each module and each agent of Flow-of-Action, with the results summarized in Table 3. When the SOP was removed, lacking domain-specific guidance, the model relied solely on its own orchestration, essentially reverting to ReAct. The significantly low accuracy underscores the crucial role of SOP. It is worth mentioning that when SOP knowledge is removed, the SOP flow becomes ineffective as well, thus removing SOP knowledge is equivalent to removing both SOP knowledge and SOP flow.

Upon removing the prompts related to the SOP flow, we noticed a significant decrease in LA, while TA remained relatively effective. This is because SOP knowledge and relevant tools were still present and could provide type information through tools like $match\_observation$ or $match\_sop$. However, the absence of the flow hindered the complete execution of the SOP, leading to the incapacity to discern location information.

The absence of the action set rendered the model unable to make correct judgments in complex and rare scenarios. However, in most cases, the model still performed adequately, resulting in a moderate decrease in effectiveness. Without the action set, the model tended to rely more on tools determined by the flow, reducing the likelihood of excessive tool invocations and thus significantly lowering APL.

At the multi-agent level, the removal of any single agent led to a certain degree of decrease in accuracy. This is attributed to the complexity of the RCA task, where having a single agent handle

all processes may lead to oversight and hallucinations. In contrast, a MAS with one main agent and multiple auxiliary agents effectively addresses this issue. The main agent can make decisions by considering the opinions of others, reducing the cognitive load and consequently achieving higher accuracy.

Regarding APL, apart from the significant impact of removing the action set, the effects of other ablations were relatively similar. This is due to the imposed limit of 20 steps to prevent unbounded loops that could render the RCA process unending.

# 4 RELATED WORK

## 4.1 TRADITIONAL METHODS

The traditional RCA methods can be categorized into four types based on the data modalities they utilize: (1) Metric-based Methods (Kocaoglu et al., 2019; Ikram et al., 2022; Li et al., 2022a; Wang et al., 2023a): These typically involve constructing bayesian causal networks or graphs using data such as Remote Procedure Call (RPC). RCA is then performed through techniques like random walks or counterfactual analysis on these networks or graphs. (2) Log-based Methods (Amar & Rigby, 2019; Rosenberg & Moonen, 2020): These focus on analyzing log data, such as examining changes in log templates or extracting specific keywords. These approaches aim to detect anomalies and simultaneously identify root causes. (3) Trace-based Methods (Yu et al., 2021; Liu et al., 2020): These methods identify root causes by observing changes in trace patterns. For instance, MicroRank (Yu et al., 2021) compares trace distributions before and after a failure to calculate anomaly scores. SparseRCA (Yao et al., 2024b) employs historical data to train pattern recognition models for root cause identification. (4) Multi-modal Methods (Yao et al., 2024a; Yu et al., 2023): These approaches posit that each data modality can, to some degree, reflect the root cause. It typically involves converting all data modalities into events or alerts, constructing a graph, and applying algorithms like PageRank (Page, 1999) to localize the root cause.

## 4.2 LLM-BASED METHODS

Due to its powerful natural language analysis and reasoning capabilities, LLMs have gradually been applied in RCA. Chen et al. (2024b) utilizes LLMs for summarization and recalls historically similar incidents to deduce the root cause of current issues. RCAgent (Wang et al., 2023b) leverages code and log data to construct an agent based on ReAct for automated orchestration in root cause localization. mABC (Zhang et al., 2024) adopts a more rational multi-agent framework and introduces a blockchain-based voting mechanism among agents. D-Bot (Zhou et al., 2024) similarly employs a multi-agent framework, refining tool selection and knowledge structure. However, these methods are predominantly designed for specific scenarios such as databases, incorporating many context-specific elements like agent categories, thereby limiting their generalizability and transferability.

# 5 CONCLUSION

The occurrence of frequent incidents necessitates RCA for swift issue resolution. Applying LLM agents in RCA presents numerous challenges. To address these challenges, we propose Flow-of-Action, a novel SOP-enhanced MAS. Flow-of-Action effectively leverages SOP knowledge by designing the SOP flow to alleviate hallucinations in the orchestration process. The action set mechanism efficiently tackles the challenge of selecting appropriate actions in the face of diverse observations. By employing a main agent supported by multiple auxiliary agents, Flow-of-Action further refines the delineation of responsibilities among agents, thereby enhancing the overall accuracy. Experimental results demonstrate the efficacy of Flow-of-Action in RCA.

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

## A  REPRODUCIBILITY

Regarding the issue of reproducibility, we will provide detailed implementation details and examples below. As for the code, many of the tools are application-specific, making it both challenging and of limited value to make them publicly available. However, we plan to integrate the entire framework into a package for public use in future work. Concerning the data, microservice framework, and monitoring system that we have developed, we will consider releasing them after the anonymization process has been completed.

## B  IMPLEMENTATION DETAILS

### B.1  PROMPT OF MULIT-AGENT SYSTEM

**Prompt of JudgeAgent**

```
Currently, an anomaly happened in Kubernetes system. The following
 is the history of the diagnose history between a user and a
aisstant:

``````History Begin``````
${diagnose_history}
``````History End``````

## Defination of Root Cause
A root cause generally consists of the following three parts, only
 when all three parts are correctly found can the root cause be
found. The following are the defination of three parts:
1. Location (which pod, service usaually isn't a correct location.
 If all three pods(-0,-1 and -2) of a service are anomalous, then
the location is service name).
2. Anomaly type. All types: pod failure, network loss, network
corrupt, network delay, network duplicate, network partition,
network bandwidth, cpu stress, memory stress. Anything outside of
these types is not a correct type.
3. Anomaly reason (Metric increase, decrease, high metric or low
metric isn't an correct anomaly reason).

## The following are some correct and incorrect root causes:
1. Location: adservice-1, Anomaly type: network loss, Anomaly
reason: context cancelled.  [Incorrect, since adservice is a
service, not a pod]
2. Location: adservice-0, Anomaly type: network delay, Anomaly
reason: rtt decrease.  [Incorrect, since metric status isn't a
correct anomaly resaon]
3. Location: adservice-0, Anomaly type: pod failure, Anomaly
reason: TCP failed to xxx.xx.xxx.xx.  [Correct]

Task
Your task is to judge whether the root cause has been found
correctly.

For example:
{"judgement": "No", "analysis": "Root cause hasn't been found
since the anomaly reason isn't sure ..."}
```

Remember to respond using json string format (can be directly
parsed by json.loads) with two json key (judgement and analysis)
without any other words.

**Prompt of ObAgent**

The following are some historical fault manifestations and their
fault types. Now there is a new anomaly.

``````History Faults``````
${history_faults}
``````History Faults End``````

``````New Faults``````
${new_fault}
``````New Faults End``````

Your task is to determine the type of this new fault based on the
manifestations of these faults and this new fault. You can do this
 task with the following steps:
1. Find the differences of the historical anomaly manifestations.
2. Decide the type of the new fault according to the differences.

Simply give the type and a simple analysis (no more than 100 words
).

For example:
The fault class is likely to be ...
The fault class is uncertain since it's not similar to all the
history manifestations...

**Prompt of ActionAgent**

According to the above chat history, give ${action_set_num}
suggested actions using json format.

# Some rules for suggesting actions:
1. When last action is run_sop and some error happened, you should
 probably suggest generate_sop_code to regenerate the correct code
 and choose the correct parameters.
2. When last action is match_standard_operation_procedure and find
 none reasonable sop, you should suggest generate_sop to generate
new sop.
3. When last action is match_standard_operation_procedure and find
 a matched sop, you should suggest generate_sop_code to generate
the code.
4. When last action is match_observation and find the anomaly type
 is uncertain or ambiguous, you should suggest
whether_is_abnormal_metric or collect_trace to get more
information.
5. When last action is match_standard_operation_procedure and find
 none sop, you should suggest it again but use the right
parameters.
6. When last action is generate_sop and get the new sop, you
should suggest generate_sop_code to generate the code of the sop
and then use run_sop to run the code.
7. Try to use as many tools as possible. If possible, don't call
the same tool with the same argument more than once!

```
8. Don't guess, for example, the name of a service or the name of
a metric.
9. For an SOP, if it is successfully executed with
generate_sop_code run_sop and the correct observation is obtained,
 then the SOP should not be executed again in a short period of
time.

Respond with a json string that can be directly parsed by json.
loads, the json keys are the {action_set_num} suggested action
names, the json values are suggested reason (no more than 20 words
).

Remember respond with a json string that can be directly parsed by
 json.loads without any other words.
```

**Prompt of MainAgent**

```
You are in a company whose Kubernetes system meet an anomaly. The
anomaly alert info is:
${alert_info}

Your task is to find the root cause of the anomaly, you can take
many steps to do the task. The following are some rules that you
should obey.

# Rules and Format Instructions for Analysis
When you are asked to give some analysis, just give some an
analysis based on the chat history especially the last observation
.

# Rules and Format Instructions for Tool Using
If at the beginning and last action doesn't exist:
    next action should be match_standard_operation_procedure
If last action == match_standard_operation_procedure:
    last observations are all matched SOPs
    next action should be generate_sop_code # Parameters:
cause_name of the SOP document should be the unexcuted SOP with
higher score, you shouldn't excute one SOP twice. If one SOP has
been excuted already, choose another one.
    If no SOPs matched or the SOPs are not relevant:
        next action should be generate_sop
elif last action == generate_sop_code:
    last observations are code
    last action should be run_sop
elif last action == run_sop:
    last observations are result after running code
    if some error happenend:
        next action should be generate_sop_code # regenerate the
right code
    else:
        next action should be match_observation # Parameters: the
query should be the whole original observation without any delete
elif last action == match_observation:
    last observations are possible anomaly class
    next action should be match_standard_operation_procedure #
match SOP of the possible anomaly class
elif last action == generate_sop:
    last observation is the new SOPs you got.
    next action should be generate_sop_code to generate the code
```

If three part of the root cause (Location (which pod, service isn't a right location), anomaly type (All types: pod failure, network loss, network corrupt, network delay, network duplicate, network partition, network bandwidth, cpu stress, memory stress) and anomaly reason (high or low metric isn't a correct reason)) have been correctly founed:
    next action should be Speak # root cause is location, anomaly type and anomaly reason

# Some Other Rules
1. You shouldn't judge the anomaly class by the metric, for example, rtt anomaly doesn't means network delay.
2. Don't make wild guesses, try to rely on evidence.
3. Don't call a tool repeatedly with the same arguments

Based on the above diagnose history, ${agent_name}, what will you do?

**Prompt of CodeAgent**

Currently, one user are diagnosing a fault, and the user is continuously interacting with the assistant. The following is the diagnose history:

``````History Begin``````
${diagnose_history}
``````History End``````

At the end of history, the assistant want to translate an SOP into python code using generate_sop_code. The SOP he choose is as follows:

SOP Name: ${sop_name}
${sop}

Your task is to translate the above choosed SOP into python code according to all the information you have.

There are some rules you should obey when you generate the code.
1. If the value of the variable you define can be analyzed through the diagnose history, you should assign it as much as possible.
2. Your code should strictly follows the SOP steps which the assistant chooses.
3. The end of the code should be answer = ...
4. The code needs to strictly follow Python syntax.
5. All the functions return type is str, so the last line of the code is answer = ... + ...

For example:
start_time = ... # find the time in diagnose history
end_time = ... # find the time in diagnose history
rtt_status = whether_is_abnormal_metric(start_time, end_time, 'rtt')
...
answer = rtt_status + ...

Respond with the json string format (can be directly parsed by json.loads) with key 'code' without any other words!

```
For example:
{{\n  "code": "start_time = \'2024-07-31 14:55:05.467000+00:00\'\\
nend_time = \'2024-07-31 15:00:05.467000+00:00\'"\n}}

Rmember to give me the code with the json string format!
```

## B.2   MULTIMODAL DATA MONITORING SYSTEM

We first deploy various data collection systems. For metrics, we start by deploying Prometheus, which collects architecture-level metrics, such as pod-level and node-level indicators that are generally standardized and unrelated to business logic (e.g., pod_network_transmit_packets). Additionally, we deploy DeepFlow to gather business-level metrics, such as business traffic data. For anomaly detection, we use traditional rule-based methods because they are fast and convenient.

For trace data, we deploy Jaeger to collect all trace data, where each trace represents a call chain containing multiple spans, with each span corresponding to a single call. Anomalies can occur within any span. In the current environment, detecting trace anomalies is relatively straightforward, as a span failure typically includes an associated error message. Therefore, we directly extract error messages to generate alert reports. For log data, we use Elastic for collection. Since abnormal logs usually contain specific keywords, extracting anomalies based on keywords has become widely accepted. We also adopt this keyword-based approach for log anomaly detection.

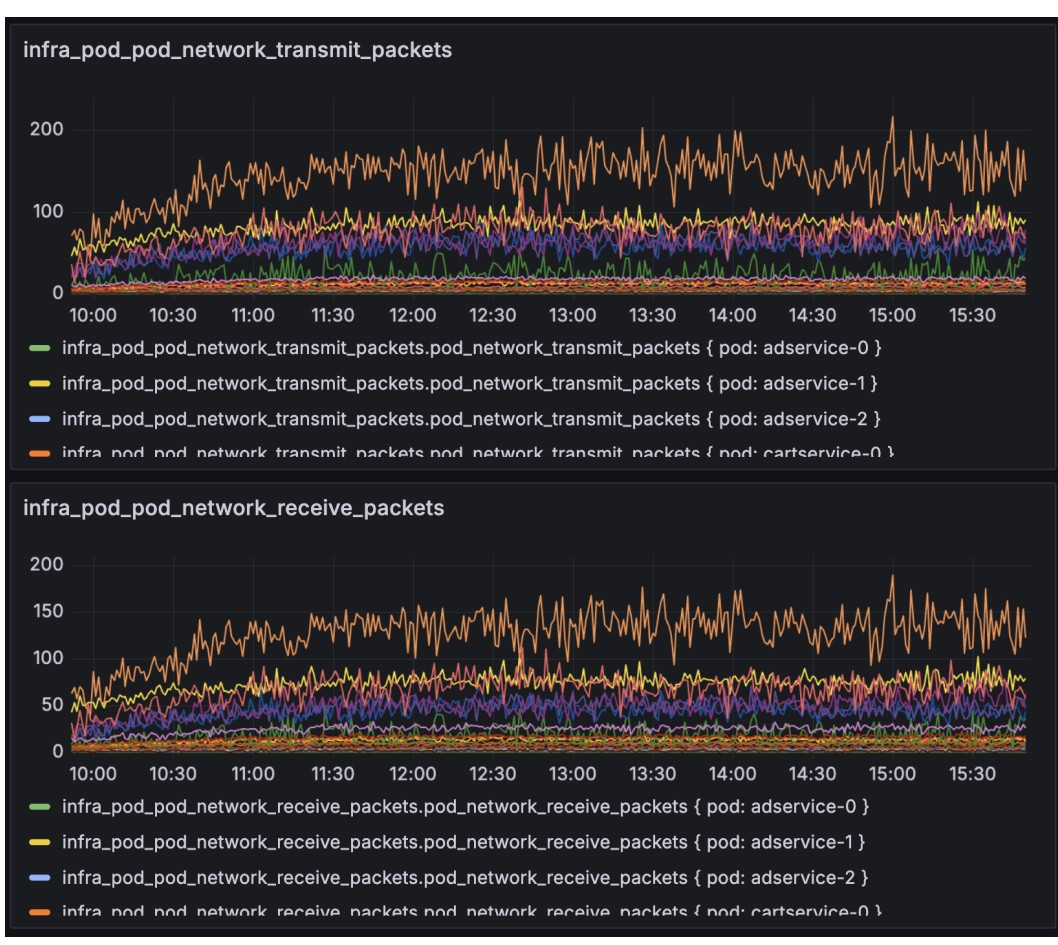

Figure 6: Prometheus Dashboard.

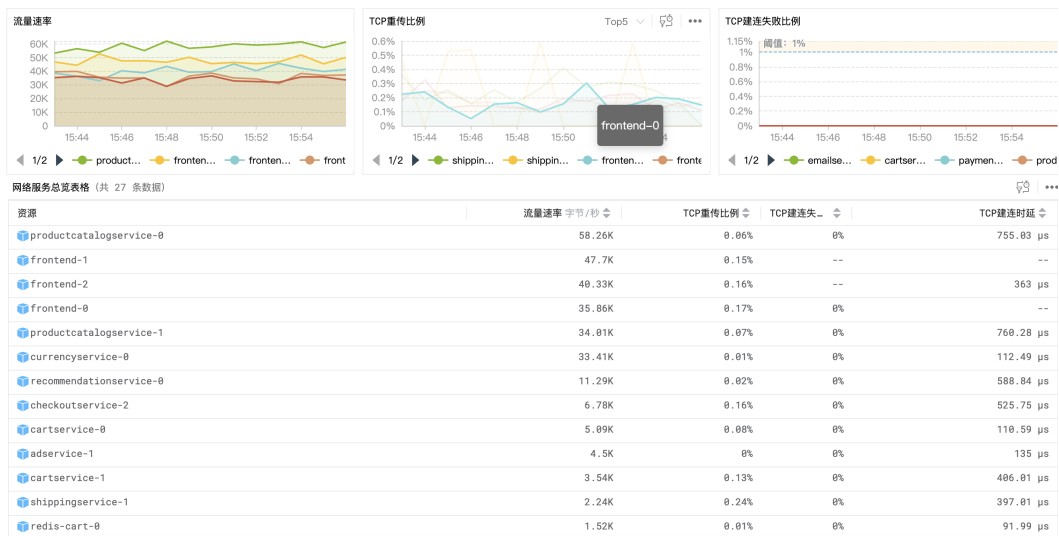

Figure 7: Deepflow Dashboard.

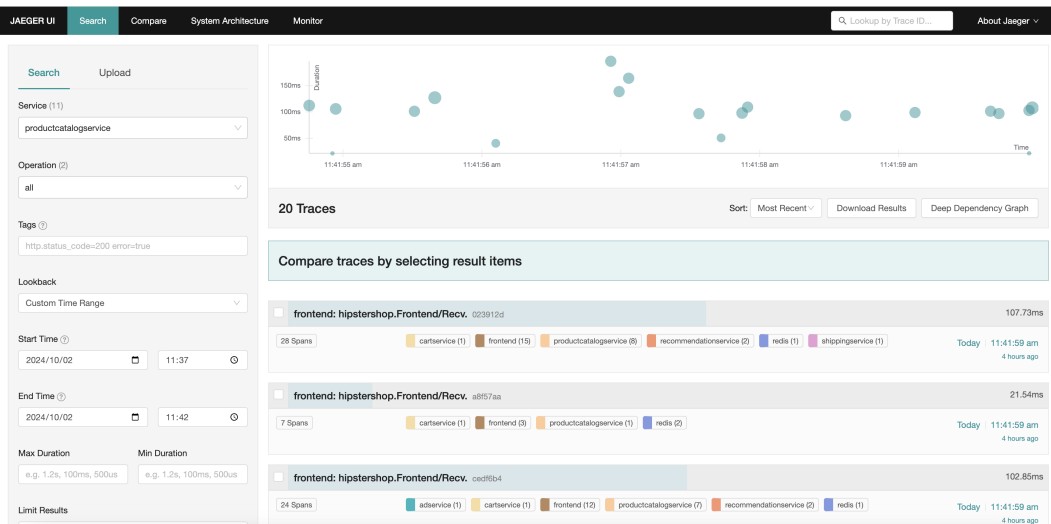

Figure 8: Jaeger Dashboard.

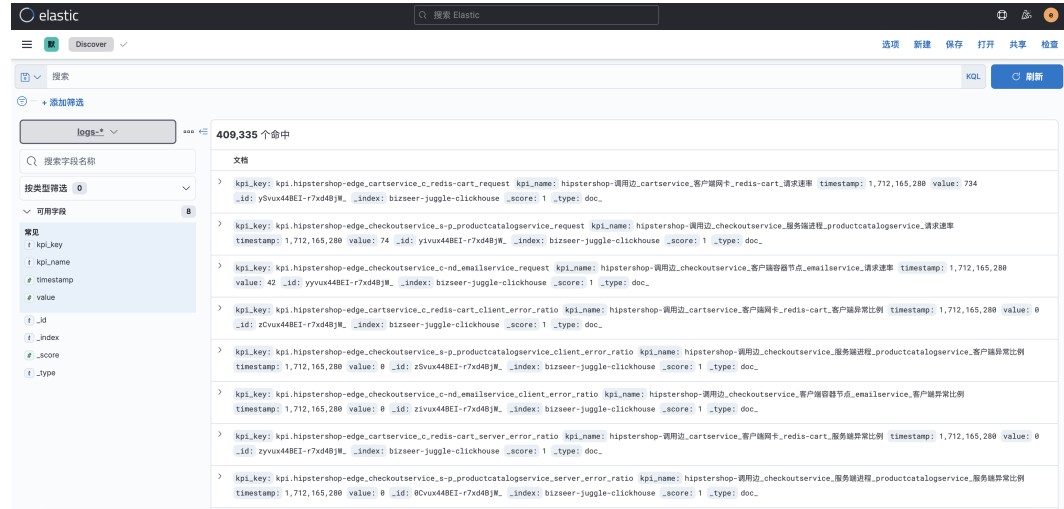

Figure 9: Elastic Dashboard.

## C  EXAMPLES OF FLOW-OF-ACTION

### C.1  EXAMPLES OF *generate_sop*

**IO Error**

```
1. get_relevant_metric: Get the relevant metrics related to the
anomaly.
2. whether_is_abnormal_metric: Check if the IO metrics are
abnormal.
3. collect_trace: Collect trace data for the anomalous service to
investigate further. (start_time, end_time, servicename)
4. The answer is the observations obtained from former steps.
```

**Node Error**

```
1. node_analyze: Analyze the status of nodes in the Kubernetes
system to identify any anomalies related to the nodes.
2. run_kubectl_command: Run the kubectl command to get services
relevant to the nodes.
3. collect_trace: Collect anomalous trace data for the services to
 gather more information about the error.
4. whether_is_abnormal_metric: Check the relevant metrics status (
such as node_cpu_usage_rate) of the nodes to determine if there
are any abnormal metrics.
5. The answer is the observations obtained from former steps.
```

### C.2  EXAMPLES OF *generate_sop_code*

**Network Partition Error**

```
start_time = '2024-09-27 20:17:52+08:00'
end_time = '2024-09-27 20:25:52+08:00'
retrans_ratio_status = whether_is_abnormal_metric(start_time,
end_time, 'retrans_ratio')
rtt_status = whether_is_abnormal_metric(start_time, end_time, 'rtt
')
tcp_establish_fail_ratio_status = whether_is_abnormal_metric(
start_time, end_time, 'tcp_establish_fail_ratio')
```

```
byte_status = whether_is_abnormal_metric(start_time, end_time, '
byte')
answer = retrans_ratio_status + ' ' + rtt_status + ' ' +
tcp_establish_fail_ratio_status + ' ' + byte_status
```

**Pod Error**

```
start_time = '2024-09-27 20:17:52+08:00'
end_time = '2024-09-27 20:25:52+08:00'
anomalous_pod = 'adservice-1'
pod_status = pod_analyze(anomalous_pod)
pod_log_status = kubectl_logs(anomalous_pod, start_time, end_time)
answer = pod_status + pod_log_status
```

## D OTHERS

Table 4: Description of Tools

| Tool | Description |
| --- | --- |
| pod_analyze | Analyzing all pods' status. |
| node_analyze | Analyzing all nodes' status. |
| service_analyze | Analyzing all services' status. |
| deployment_analyze | Analyzing all deployments' status. |
| statefulset_analyze | Analyzing all statefulsets' status. |
| run_kubectl_command | Executing kubectl commands generated by LLMs. |
| get_all_namespace | Obtaining a list of all namespaces. |
| get_relevant_metric | Obtaining relevant metric names according to query. |

Table 5: Fault Types

| Type | Description |
| --- | --- |
| CPU Stress | Generate some threads to occupy CPU resources. |
| Memory Stress | Generate some threads to occupy memory. |
| Pod Failure | Make the pod inaccessible for a period of time. |
| Network Delay | Causes network delay for a pod. |
| Network Loss | Causes packet loss in a pod's network. |
| Network Partition | Network disconnection, partition. |
| Network Duplicate | Causes a pod's network packet to be retransmitted. |
| Network Corrupt | Causes packets on a pod's network to be out of order. |
| Network Bandwidth | Limit the bandwidth of communication between nodes. |

```
# Rules and Format Instructions for Tool Using
If at the beginning and last action doesn't exist:
    next action should be match_sop
If last action == match_sop:
    last observations are all matched SOPs
    next action should be generate_sop_code # Parameters: cause_name of the SOP document should be the unexcuted SOP
with higher score, you shouldn't excute one SOP twice. If one SOP has been excuted already, choose another one.
    If no SOPs matched or the SOPs are not relevant:
        next action should be generate_sop
elif last action == generate_sop_code:
    last observations are code
    next action should be run_sop
elif last action == run_sop:
    last observations are result after running code
    if some error happenend:
        next action should be generate_sop_code # regenerate the right code
    else:
        next action should be match_observation # Parameters: the query should be the whole original observation without any
delete
elif last action == match_observation:
    last observations are possible anomaly class
    next action should be match_sop # match SOP of the possible anomaly class
elif last action == generate_sop:
    last observation is the new SOPs you got.
    next action should be generate_sop_code to generate the code
```

Figure 10: Prompt used to pass the SOP flow information to agents.

