# OpenReview forum: "Flow-of-Action: SOP Enhanced LLM-Based Multi-Agent System for Root Cause Analysis"
_ICLR.cc/2025/Conference — ICLR 2025 Conference Withdrawn Submission_

### Official Review · Reviewer_hEu6 · 2024-11-07

**Soundness:** 1
**Presentation:** 3
**Contribution:** 2
**Rating:** 3
**Confidence:** 4

**Summary:**

Root cause analysis (RCA) is a structured, step-by-step process designed for fault localization in software systems.
This paper presents a Standard Operating Procedure (SOP) enhanced Multi-Agent System (MAS) for RCA in microservice based architecture.
The authors proposed a large language model (LLM) based multi-agent system and evaluated their approach empirically
on real-world open source microsevice-based architecture RCA tools, such as
K8SGPT and HolmesGPT. The authors also developed baselines some general-purpose open-source frameworks, such as CoT, ReAct, and Reflexion.

**Strengths:**

- A RCA system for complex systems.
- Designed SOP knowledge to alleviate LLM hallucinations
- Relied on historical data to match new anomalies and identify faults
- Performance boost compared to state-of-the art methods

**Weaknesses:**

While the research direction is promising, there are several weakness.
First, in Section 2.3.1, the authors described “To enhance the precision of RCA, we have devised hierarchical
SOPs. Our objective is for the RCA process to progress from a macro to micro level, from a general
to specific perspective, mirroring real-world scenarios more closely. For instance, we first address
network issues before delving into network partition problems"


How does the SOP knowledge change when a new SOP is introduced? Does the knowledge base update every time?


Second, in Section 2.4, how do the authors confirm that the action set is comprehensive?

Third, failures and fault localization in real-world micro service-based architecture is well-studied problem in software engineering research [1-3]. The experiment design in Section 3.1.1,  specifically artificially fault injection in microservice-based architecture is not well justified.

Fourth, how the authors derived the rule-set of MainAgent on page 15 is not clear to me. As this ruleset is crucial in proposed model's performance evaluation, a systemic and well-defined approach is expected. An example of deriving a ruleset based on historical software artifacts can be found in [4].


References.
1. Meng, Y., Zhang, S., Sun, Y., Zhang, R., Hu, Z., Zhang, Y., Jia, C., Wang, Z. and Pei, D., 2020, June. Localizing failure root causes in a microservice through causality inference. In 2020 IEEE/ACM 28th International Symposium on Quality of Service (IWQoS) (pp. 1-10). IEEE.
2. Zhang, S., Xia, S., Fan, W., Shi, B., Xiong, X., Zhong, Z., Ma, M., Sun, Y. and Pei, D., 2024. Failure Diagnosis in Microservice Systems: A Comprehensive Survey and Analysis. arXiv preprint arXiv:2407.01710.
3. Jagadeesan, L.J. and Mendiratta, V.B., 2020, October. When failure is (not) an option: Reliability models for microservices architectures. In 2020 IEEE International Symposium on Software Reliability Engineering Workshops (ISSREW) (pp. 19-24). IEEE.
4. Rahman, A., Parnin, C. and Williams, L., 2019, May. The seven sins: Security smells in infrastructure as code scripts. In 2019 IEEE/ACM 41st International Conference on Software Engineering (ICSE) (pp. 164-175). IEEE.

**Questions:**

Please see weakness.

---

### Official Review · Reviewer_rnUG · 2024-11-08

**Soundness:** 2
**Presentation:** 2
**Contribution:** 2
**Rating:** 3
**Confidence:** 3

**Summary:**

The paper proposes an LLM-based multi-agent system for the root cause analysis problem. It focuses on mitigating two issues that are prevalent in the existing LLM-based RCA methods: hallucinations and the existence of multiple possible actions for a given observation. Both of them can negatively affect the accuracy and efficiency of the model performance. For the first issue, it proposes to use a Standard Operation Procedure (SOP), and some background knowledge from engineers, to guide LLMs for RCA. It further introduces a set of predefined functions called SOP flow to obtain the necessary information for harnessing SOPs. To tackle the second issue, the authors introduce an extra step in the existing paradigm to avoid immediate action selection and instead generate a reasoned action set before making the final decision on the course action. Also, the paper specifically introduces multiple different agents to take part of the process of RCA for enhancing the model performance. The experiment simulated from real-world e-commerce data shows that the proposed method outperforms its baselines. Overall, the paper needs to elaborate more on the details of the model in the main paper. Along with the lack of comparison with other non-LLM-based RCA methods, I lean more toward rejecting the paper.

**Strengths:**

- The proposed method proposes to take advantage of standard procedures available from engineers to mitigate hallucination issues in LLMs.
- The authors also propose multi-agent systems to provide a more informative diagnosis of RCA beyond identifying the root cause.
- The paper introduces a set of tools called the SOP flow to better utilize SOPs.
- The paper provides a lot of visualizations to enhance the presentation of the paper.
- The paper includes a simulation to demonstrate the advantages of the proposed method.

**Weaknesses:**

- The comparison with other existing methods for RCA is largely missing. The paper merely focuses on improving upong the ReAct framework but didn’t discuss why ReAct is preferred in the first place among all other RCD methods. It only talks about a single RCD method that uses deep learning.
- This paper extends a framework ReAct from a paper that has not been published. It becomes even more difficult to understand the novelty of this extension. Another cited paper is recently published, but there is no discussion on how this paper is compared to the cited one.
- Some parts of the methodology are not described in detail. The paper often discusses many processes only at a high level, which makes it difficult to judge the correctness of the approach-related various claims that often appear subtly in the paragraph.
- The paper advocates that the proposed method is efficient, but it is limited to the context of LLM-based RCA methods. In terms of efficiency, a comparison with other non-LLM-based methods is missing.

**Questions:**

- How good is the new SOPs generation when it is not given by the engineers?
- How do the authors evaluate the reasoned action set in order to pick the final action?
- Line 154 "This approach, however, does not consistently yield optimal results in RCA. Therefore, we have devised an innovative knowledge base model integrating SOP knowledge and historical incident knowledge.” Does the experiment support this?
- The standard operation procedure seems to be a time-consuming task for engineers to handle esepcially for a large-scale system? Do the authors take this into account for the efficiency of the proposed algorithm?
- How does the ObAgent balance how much it relies on historical context vs. the new data for RCA?
- Lines 332-333: "we have established a rule based on the flow to ensure that the action set is comprehensive and logical” can the authors elaborate on this and what the rule is exactly?
- Why is the motivation for using multiple agents? Why can’t I have a single agent to do everything?
- Line 353: "Taking additional steps and gathering more information may lead to a more accurate root cause determination” What are the additional steps?
- There are many existing RCA methods that also perform very well on the GoogleOnlineBoutique data e.g. BARO [1], why aren’t they included in the experiment for comparison? Based on my initial observations, the existing baselines in [1] seem to perform quite well. But I am not sure if the implementation is exactly the same. Can the authors give the performance of BARO under the same setup for location accuracy? I think it is a fair game to show comparison with other methods beyond LLM-based since the problem is about RCA.

Reference:

[1] Pham, Luan, Huong Ha, and Hongyu Zhang. "Root Cause Analysis for Microservice System based on Causal Inference: How Far Are We?." arXiv preprint arXiv:2408.13729 (2024).

[2] Pham, Luan, Huong Ha, and Hongyu Zhang. "Baro: Robust root cause analysis for microservices via multivariate bayesian online change point detection." Proceedings of the ACM on Software Engineering 1.FSE (2024): 2214-2237.

---

### Official Review · Reviewer_G1jn · 2024-11-08

**Soundness:** 3
**Presentation:** 2
**Contribution:** 2
**Rating:** 5
**Confidence:** 4

**Summary:**

The paper proposed an agentic solution for root cause analysis of cloud microservices. As the naïve ReAct agents are prone to making mistakes and hallucinations, a principal way of generating SOPs and constraining the agent to follow the SOP plans are proposed for guiding RCA process to right directions. SOP flow can generate automated codes for executing different relevant tools which reduces hallucinations. Furthermore, a multi-agent framework is proposed with judge and observer agent to streamline the overall execution. Experimental results on a simulated production environment (synthetic faults injected using chaos engineering) demonstrate that the proposed framework can achieve significant boost in accuracy over ReAct and other agentic solutions.

**Strengths:**

The paper solves a practically important and challenging problem in cloud service management that can have significant impact in industries. Root cause analysis in microservices requires significant amount of domain knowledge and multiple sources of dataset (including logs, traces, metrics, diagnostic documents). Automating the RCA workflow using LLM agents (with high accuracy) can significantly reduce the manual efforts from engineers. While ReAct based agentic solutions by leveraging troubleshooting guides have been proposed earlier, efficacy of these agents is still poor due to LLM hallucinations. This paper takes a significant leap forward by organizing the RCA workflow with SOPs and automatically generating executable codes from SOPs. Moreover, multi-agent architecture proposed in the paper is a step in right direction that include supportive observer and judge agents. Experiments are conducted on a simulated production environment from which SOPs can be generated accurately, and the results are promising -- demonstrating a significant boost in accuracy over existing ReAct agents.

**Weaknesses:**

Although I appreciate the multi-agent and SOP based solution framework and performance gain in simulated environment, I have a few concerns:
1. The central contribution of the work lies in creating structured SOPs. This is not a challenging task when operating on a simulated environment where faults are artificially injected, and the nature of the faults are known. However, in real production environments, where thousands of inter-dependent microservices are continuously running, identifying the nature of the fault and localization itself is a challenging task. I am not fully sure how these structured SOPs can be prepared in a real-world setting. Is there a way to automate the SOP creation process given the nature and dependency of the services? If SOPs need to be prepared by the service owners, then it is a strong assumption and may not be a practical solution.
2. While multi-agent framework is proposed as a key contribution, several details regarding this is missing. What multi-agent framework is used in the experiment and how these agents are communicating among each other needs to be clarified in detail. Does the execution of agents follow a sequential static flow, or they run asynchronously and somehow manage their communication flow. Please clarify.
3. Experiments are conducted in a simulation environment where faults are injected synthetically. Therefore, the actual performance of the proposed framework in a real-world setting is still questionable as the real situation is much unorganized and chaotic.
4. Lastly, several details on experiments are missing in the paper which raises concerns about reproducibility. Are you planning to opensource the code and simulated data?

**Questions:**

1. Have you compared the performance with existing RCA agent solution [1] that leverages trouble shooting documents?
2. As LLM agents' execution and outputs are non-deterministic, have you done any sensitivity analysis to check if the performance remains unchanged over multiple iterations?
3. How to automatically create the SOPs in a real-world production environment? Do you expect service owners to create these documentations?
4. Have you performed any analysis on accuracy of SOP code generation from given SOPs? What percentage of cases, the generated codes are accurate?
5. How is the domain adaptation issue handled? As LLMs are not trained with production data, it is highly possible that the LLM generated codes might miss nuances of the services. Have you done any finetuning with RCA domain data?

[1] Roy, D., Zhang, X., Bhave, R., Bansal, C., Las-Casas, P., Fonseca, R. and Rajmohan, S., 2024, July. Exploring llm-based agents for root cause analysis. In Companion Proceedings of the 32nd ACM International Conference on the Foundations of Software Engineering (pp. 208-219).

---

### Note · Authors · 2024-11-29

I have read and agree with the venue's withdrawal policy on behalf of myself and my co-authors.